# Effects of High Intensity Interval Training Rehabilitation Protocol after an Acute Coronary Syndrome on Myocardial Work and Atrial Strain

**DOI:** 10.3390/medicina58030453

**Published:** 2022-03-21

**Authors:** Antonello D’Andrea, Andreina Carbone, Federica Ilardi, Mario Pacileo, Cristina Savarese, Simona Sperlongano, Marco Di Maio, Francesco Giallauria, Vincenzo Russo, Eduardo Bossone, Eugenio Picano

**Affiliations:** 1Unit of Cardiology, Department of Traslational Medical Sciences, University of Campania “Luigi Vanvitelli”, Monaldi Hospital, 80131 Naples, Italy; andr.carbone@gmail.com (A.C.); sperlongano.simona@gmail.com (S.S.); v.p.russo@libero.it (V.R.); 2Unit of Cardiology and Intensive Coronary Care, “Umberto I” Hospital, 84014 Nocera Inferiore, Italy; m.pacileo@aslsalerno.it (M.P.); c.savarese@aslsalerno.it (C.S.); 3Department of Translational Medical Sciences, University of Naples Federico II, 80131 Naples, Italy; fedeilardi@gmail.com (F.I.); francesco.giallauria@unina.it (F.G.); 4Unit of Cardiology, “Eboli Hospital (ASL Salerno)”, 84025 Eboli, Italy; marcodimaio88@gmail.com; 5Cardiac Rehabilitation Unit, Cardarelli Hospital, 80131 Naples, Italy; ebossone@hotmail.com; 6Institute of Clinical Physiology, CNR, 56127 Pisa, Italy; picano@ifc.cnr.it

**Keywords:** strain, myocardial work, rehabilitation, cardiopulmonary test, acute coronary syndrome, chronic coronary syndrome, echocardiography, diastole, left atrium

## Abstract

*Background and**Objectives***:** Current guidelines on cardiac rehabilitation (CR) suggest moderate-intensity physical activity after acute coronary syndrome (ACS). Recent report have shown that high-intensity interval training (HIIT) could be more effective than moderate-intensity continuous training (MCT) in improving cardiac performance. Our aim was to analyze the effects of HIIT protocol after ACS on advanced echocardiographic parameters of myocardial function. *Materials and Methods***:** In total, 75 patients with recent ACS, with or without ST segment elevation, were enrolled and compared with a control group of 50 age- and sex-comparable healthy subjects. Patients were randomized to perform a MCT training or HIIT-based rehabilitation program. A complete echocardiographic evaluation, including left ventricular (LV) and left atrial (LA) global longitudinal strain (GLS) and myocardial work (MW) through speckle-tracking analysis, was performed for all patients, before and after cardiac rehabilitation training. A cardiopulmonary exercise testing (CPET) was also performed at the end of the rehabilitation program. *Results:* Patients who followed the HIIT rehabilitation program showed improved LV diastolic function compared to the MCT group (E/e’: 3.4 ± 3.1 vs. 6.4 ± 2.8, respectively, *p* < 0.01). Similarly, LV systolic function showed significant improvement in the group of patients performing HIIT (ejection fraction: 53.1 ± 6.4 vs. 52.3 ± 5.4%, *p* < 0.01; GLS: −17.8 ± 3.8 vs. −15.4 ± 4.3, *p* < 0.01). In addition, LA strain was improved. MW efficiency was also increased in the HIIT group (91.1 ± 3.3 vs. 87.4 ± 4.1%, *p* < 0.01), and was closely related to peak effort measurements expressed in peak VO_2_ by CPET. *Conclusions***:** In patients with recent ACS, the HIIT rehabilitation program determined reverse cardiac remodeling, with the improvement of diastolic and systolic function, assessed by standard echocardiography. In addition, cardiac deformation index as GLS, LA strain and MW efficiency improved significantly after HIIT, and were associated with functional capacity during effort.

## 1. Introduction

Cardiac rehabilitation (CR) is a complex intervention offered to patients usually after myocardial infarction, cardiac surgery or with chronic heart failure [1]. It includes lifestyle behaviour changes, psychosocial aid and exercise programs [1,2]. In particular, the importance of starting an aerobic exercise-based secondary prevention program has been confirmed by its favourable effects on left ventricular (LV) remodelling, and by the improvement in the mortality and morbidity of patients with stable coronary artery disease (CAD) and after an acute coronary syndrome (ACS) [3,4]. Current guidelines on CR suggest moderate- or moderate-to-high-intensity physical activity, according to the individual patient and disease characteristics [1].

On the other hand, high-intensity interval training (HIIT) would seem to be more effective than moderate-intensity continuous training (MCT) in improving cardiac performance and function [1]. HIIT generally consists of exercise sessions with a muscular warm-up in steady-state mode for about 5 min, followed by high-intensity exercises repetitions (at 80–90% of maximum heart rate) separated by exercises of medium intensity for recovery [1]. However, few reports have documented the effects of HIIT on diastolic function, and on LV and left atrial (LA) deformation parameters such as Global Longitudinal Strain (GLS).

Recently, a new echocardiographic index for the evaluation of LV deformation and function was introduced, namely myocardial work (MW). MW considers cardiac afterload in the strain analysis through the relation with non-invasive LV pressure [5]. This technique could improve the study of LV function and deformation in relation to disparate overloading conditions, as can be seen in CR patients.

Therefore, the purpose of this study was to compare the effects of two different cardiac rehabilitation protocols, HIIT vs. MCT, on cardiac remodelling and deformation indices in patients with recent ACS.

## 2. Methods

### 2.1. Study Population 

Our prospective study was based on CONSORT guidelines as a randomized controlled trial. Eighty patients with ACS, with or without ST segment elevation and undergoing complete percutaneous coronary revascularization, were enrolled within 30 days of discharge from the Intensive Coronary Care Unit of Umberto I° Hospital (Nocera Inferiore—Salerno), between September 2020 and September 2021 (NCT04511181). 

Exclusion criteria were the following: ischemic or hemodynamic instability, severe valvular diseases, major arrhythmias not well controlled, symptomatic peripheral arterial disease, any contraindication to physical activity (i.e., gonarthrosis), inability to exercise, poor acoustic window. According to these criteria, 5 patients were excluded (Figure 1).

Since the study protocol was designed for parallel groups (Figure 1), a control group of 50 age- and sex-matched healthy subjects was also included. Control subjects were excluded if they had: (1) arterial systemic hypertension, (2) stable coronary artery disease or previous acute coronary syndrome, (3) cardiomyopathy and/or a genetic heart condition, (4) congenital heart disease, (5)(4) valvular regurgitation of a higher degree than mild and/or valvular stenosis of any degree, (6) any previous cardiothoracic or vascular surgery or interventional percutaneous procedure, (7) any cardiac treatment, (8) previous cardioembolic stroke, including transient ischaemic attacks.

All patients of the study were enrolled from the population of Nocera Inferiore (Salerno), with comparable social, economic, cultural and environmental settings.

The study was approved by the Institution’s ethics committee (protocol number 121-2021), and each participant provided informed consent.

### 2.2. Study Protocol

All subjects enrolled underwent a comprehensive clinical and laboratory evaluation. Electrocardiogram (ECG), standard colour Doppler echocardiography, LV 2D speckle-tracking derived GLS and MW analysis were also performed at rest. They were randomized by a computerised random number generator to follow the MCT (40 patients) or HIIT-based (35 patients) rehabilitation program, lasting 8 weeks for a total of 16 rehabilitation sessions (Figure 1).

According to recent guidelines, HIIT consists of short periods of intense exercise (i.e., ≥85% VO_2_ peak or ≥85% heart rate reserve (HRR) or ≥90% heart rate (HR) peak) interspersed with lower level exercise, whereas MCT is a form of exercise performed with continuous intensity (i.e., 50–75% VO_2_ peak or 50–75% HRR or 50–80% HR peak) (1). After the rehabilitation treatment cycle, all patients were re-subjected to the same clinical, ECG and echocardiographic evaluation, and to a cardiopulmonary exercise testing (CPET). Both the patients and the echocardiographic examiners were blinded to the rehabilitation program.

### 2.3. Standard Doppler Transthoracic Echocardiography

A standard transthoracic echocardiography was performed using the ultrasound machine Vivid E85 (GE Ultrasound, Milwaukee, WI, USA).

LV systolic function indices were assessed, including LV ejection fraction (LVEF) with the biplane Simpson method, and the stroke volume from the LV outflow tract (LVOT) diameter and LVOT time-velocity integral. LV mass and LV mass index were calculated. LV hypertrophy was defined for a LVMI > 115 g/m^2^ for men and >95 g/m^2^ for women. Left atrial volume index (LAVI) was calculated with area-length formula. In addition, LV diastolic function was assessed (E and A peak velocities (m/s), E/A ratio, E-wave deceleration time (ms), isovolumic relaxation time (IVRT), the early diastolic septal and lateral velocities e’and the average E/e’ ratio). Peak tricuspid regurgitant velocity (TRV), systolic trans-tricuspid gradient and pulmonary artery systolic pressure (PASP) were calculated. B-lines were assessed by lung ultrasound with the 4-site scan, from mid-axillary to mid-clavicular lines on the third intercostal space, and each site was scored from 0 = A-lines to 10 = white lung.

### 2.4. Two-Dimensional Speckle Tracking Echocardiography (2D-STE)

*LV Strain.* LV GLS was assessed by the 2D STE technique with the software EchoPAC Version 202 (GE Vingmed Ultrasound, Norway). Adequate loops from the apical forth, second, and third chamber, were analysed, manually tracing the endocardial borders in the end-systolic frame of each cardiac cycle and identifying the timing of the aortic valve closure. If tracking was accurate, then the software generated a region of interest (ROI) of the entire myocardial thickness, and calculated the segmental (6 segments) and GLS for each loop. The bull’s eye summarizes the regional and GLS obtained for each cardiac segment.

*LA strain.* An ROI was traced on the endocardial border of the LA (in systole) in the apical four-chamber view. Then, an epicardial border was automatically generated. The final ROI included the entire LA myocardial wall. LA longitudinal strain myocardial thickening was indicated by a peak positive value, color-coded as red; LA myocardial thinning was characterized by a peak negative value, color-coded as blue. LA strain peak represents an index of LA passive deformation and was calculated in the middle segment of LA lateral wall.

### 2.5. MW analysis

MW was estimated with a specific software. Non-invasive arterial pressure was measured (average of three measurements) before the echocardiography and introduced in the software. The software created a LV pressure-strain loop, and the area within the loop was used as index of MW for each cardiac segment and global. Constructive work (CW, effective work performed during the systolic shortening), wasted work (WW, work did not contribute to LV ejection) and myocardial work efficiency (MWE = CW/(CW + WW) were also assessed. A bull’s eye with the segmental and global MW, CW, WW and MWE values was provided (Figure 2).

### 2.6. Cardiopulmonary Exercise Testing

Subjects underwent a cardiopulmonary exercise test after the rehabilitation protocols using an upright cycle ergometer (Ergoselect 100/200 K, Ergoline GmbH, Bitz, Germany). Respiratory gas analysis was performed with Quark PTF Ergo Cosmed (COSMED srl., Rome, Italy) with assessment of oxygen uptake (VO_2_(L/min)). Peak VO_2_ was indexed for body weight (mL/min per kg classified with Weber scale).

### 2.7. Statistical Analysis

Continuous variables were reported as median (I, III quartile) or means ± standard deviation (SD). To analyse differences within and between groups, paired and unpaired *t*-test were performed, respectively. The statistical significance was defined as a two-sided *p* value < 0.01. Intra-observer variability and inter-observer variability were calculated with the coefficient of variation (COV) and by Bland–Altman analysis.

Linear regression analysis and Pearson’s partial correlation test were performed to assess relations between echocardiographic and clinical data. The multivariate analysis, with an interactive stepwise model, was performed to identify the independent predictors of exercise capacity.

Statistical analyses were performed by SPSS for Windows release 21.0 (IBM **SPSS** Statistics for Windows, Chicago, IL, USA).

## 3. Results

The final population included 125 patients: 75 with recent ACS, and 50 age and sex-comparable healthy controls. Matching between the two groups regarding basic features is provided in Table 1. The average age of ischemic patients was 62.3 ± 8.3 years, and the majority of them were male, who had a body mass index (BMI) within the overweight range and a history of diabetes mellitus, tobacco smoking and dyslipidaemia. According to the presence of recent ACS without clinical signs of heart failure and to the mild impairment of LV contractile function, rates of therapy with diuretics, with an angiotensin-converting enzyme inhibitor or angiotensin-receptor blocker and beta-blockers, were in accordance with current guidelines.

An overview of the two populations echocardiographic parameters is shown in Table 2. Wall thicknesses, LV diameters and volumes were increased in the ischemic group compared to the control one. Conversely, a significant reduction of LA and LV systolic function indices (LVEF, GLS, MW efficiency) was observed, with a prominent increase of myocardial WW. Similarly, LV diastolic function was impaired in the group of ischemic patients, with increased LV filling pressures (E/e’ ratio).

In addition, right ventricular (RV) systolic function, assessed by traditional echocardiographic indices such as TAPSE and S’ wave rate, was impaired in ACS patients. Of note, a certain degree of pulmonary congestion was observed in the ischemic patients’ group, as evidenced by the finding of B- lines on chest ultrasound.

At the echocardiographic evaluation after CR protocols, ischemic patients showed a statistically significant improvement in LA deformation and in both LV systolic and diastolic function compared to baseline, without changes in dimensional parameters but associated with a significant reduction in lung congestion. Conversely, RV systolic functional parameters did not change significantly after the rehabilitation programs (Table 3).

When comparing the effects of the two rehabilitation protocols, patients rehabilitated by HIIT showed better improvement in diastolic function compared to rehabilitated by the MCT protocol. In addition, lower values of PAPs were detected in the HIIT group. Similarly, LA and LV contractile function showed significantly higher indices in the group of patients performing HIIT, with increased MWE (91.1 ± 3.3% vs. 87.4 ± 4.1%, *p* < 0.01), and with a significant WW reduction (9.9 ± 4.4 in HIIT group vs. 12.6 ± 3.3 in MCT group, *p* < 0.01) in the HIIT group (Table 4).

Univariate analysis showed independent associations of baseline GLS, MWE and LA strain, with LV E/e’ ratio, peak VO_2_ and B-lines during effort. These relations were stronger than the association of LVEF with the same parameters (Table 5).

LV MWE at rest was closely associated to peak VO_2_ (beta: 0.50; *p* < 0.001), LV E/e’ (beta—0.52, *p* < 0.001) and to the number of B-lines during effort (beta: −0.36; *p* < 0.01) at multivariate analysis.

### Analysis of Intra and Inter-Observer Variability

Intra-observer variability: COV: LV MWE: 5.33 (intra-class correlation, ICC 0.72); Bland–Altman analysis: LV MWE (95% confidence interval, CI ± 1.6; percent error 3.3%). Inter-observer variability: COV: LV MWE: 7.26 (ICC 0.76); Bland–Altman analysis: LV MWE (95% CI ± 1.8; percent error 3.7%).

## 4. Discussion

The main findings of our study are as followers: (i) In patients with recent ACS, HIIT rehabilitation protocol showed a significant improvement of both diastolic and systolic function compared to MCT protocol. (ii) Functional improvement in patients undergoing HIIT protocol was mainly detected by LA strain, GLS and MWE increase and WW reduction. (iii) LV MWE at rest was strongly associated with measurements at peak effort by CPET, expressed in peak VO_2_, LV diastolic function and pulmonary congestion.

### 4.1. Effects of Cardiac Rehabilitation on Myocardial Function

CR has a primary role in secondary prevention of cardiovascular conditions, involving lifestyle behaviour changes, psychological support and exercise training [1]. International societies of cardiology and preventive medicine support moderate- to vigorous-intensity exercise during CR, whereas in Australia, New Zealand, Japan and the United Kingdom, lower-intensity exercise is favoured [6,7,8].

Previous studies have clearly shown several benefits in patients receiving CR. In particular, HIIT protocol lasting more than 6 weeks appears to be more effective than MCT in improving cardio-respiratory performance in ACS patients, although prognostic information about this type of training cannot be provided [4]. In particular, in patients with CAD, HIIT protocol significantly improved VO_2_ at ventilatory threshold, LV size, cardiac contractility and endothelial function compared to MCT [3,4,9].

Whereas data on cardiorespiratory fitness improvement related to HIIT are well established, the effects of this protocol on cardiac deformation in patients with recent ACS are still little known.

Several studies have demonstrated the accuracy of GLS in detecting the LV risk area, its correlation with scar extension by cardiac magnetic resonance [10,11] and its role in predicting LV remodelling after percutaneous coronary intervention [12,13].

In addition, in ischemic patients undergoing a CR program, STE seems to provide a better quantification of LV reverse remodelling after myocardial infarction (MI), as traditional echocardiographic indexes, such as LVEF, are less sensitive for the study of subclinical contractility changes [14]. In a population of 36 MI, no changes in LV GLS were observed after 10 weeks of CR, but a significant reduction of twist and twist velocity were considered early signs of reverse LV functional remodelling and improved functional reserve [15]. Likewise Angadi et al. [16] observed a significant improvement in RV, but not LV or GLS, in a group of patients with heart failure and preserved LVEF after 4 weeks of HIIT. Conversely, Malfatto et al. [17] reported an increase of GLS in 34 MI patients undergoing CR compared to patients with MI without CR that persisted after 6 months.

### 4.2. Uniqueness of the Present Study

Our study confirmed, in a larger population of patients undergoing CR after ACS, a significant improvement of both LA and LV systolic function after 8 weeks of training, detected by an increase of LV GLS and of LA strain. This improvement in myocardial longitudinal function was even more consistent in patients following HIIT compared to MCT.

Furthermore, we provided more insight in an analysis of myocardial recovery post-CR, investigating, for the first time, the effects of two different rehabilitation programs on myocardial performance measured by LA strain and non-invasive MW indices.

MW is an innovative echocardiographic tool for the evaluation of LV deformation through the simultaneous analysis of myocardial strain and non-invasive LV pressure. Its reliability in detecting early, subclinical myocardial damage has been demonstrated in several clinical settings, also showing, at times, its superiority over GLS and other traditional indices of systolic function [5]. In non-ST-segment-elevation acute coronary syndrome (NSTE-ACS) patients with normal LVEF and systolic function, indeed, MWE was more affordable than segmental longitudinal strain to identify LV risk areas and predict obstructive coronary artery stenosis [5]. Lustosa [18] and colleagues demonstrated that 3 months after acute ST-elevation myocardial infarction (STEMI), patients with LV remodelling had more impaired global MW index, constructive work and MW efficiency, with significantly increased WW compared to patients without remodelling. In a study enrolling 93 patients with acute STEMI, constructive MW was able to independently predict segmental and global LV recovery better than other MW indices and GLS [19].

In our study population, including both STEMI and NSTEMI patients, we demonstrated that MW, in particular WW and MWE, are sensible markers of myocardial systolic function, being more impaired than controls early after ACS and significantly improving after CR. We further characterized the extent of functional recovery comparing MW indices in two different CR protocols, demonstrating higher improvement after HIIT protocol compared with MCT. In addition, given its independent association with peak effort capacity, diastolic function and signs of pulmonary congestions, both LA strain and MWE seem to be reliable tools for the study of cardiac function in patients with CAD undergoing to CR training. Thus, a complete strain study could have a key role in helping clinicians with the management of patients with CAD undergoing CR.

### 4.3. Study Limitations

The sample size is too small to give definitive conclusions concerning the myocardial deformation and work in ischemic patients. Furthermore, the group of ischemic patients is heterogenous, with different types of ACS, as well as numbers and characteristics of coronary lesions. The results of our study are applicable only in CAD patients capable of exercising.

## 5. Conclusions

CR training after ACS allows for the improvement several echocardiographic parameters, resulting in a greater functional performance of the patients and higher quality of life, regardless of the type of exercise performed. In our population of ischemic patients, HIIT showed a greater anti-remodelling effectiveness than the MCT protocols, improving not only the standard echocardiographic parameters but also the deformation index, such as LA strain, LV GLS and MWE, a LV load-independent parameter that might represent a better expression of the contractile efficiency of the LV myocardium.

## Figures and Tables

**Figure 1 medicina-58-00453-f001:**
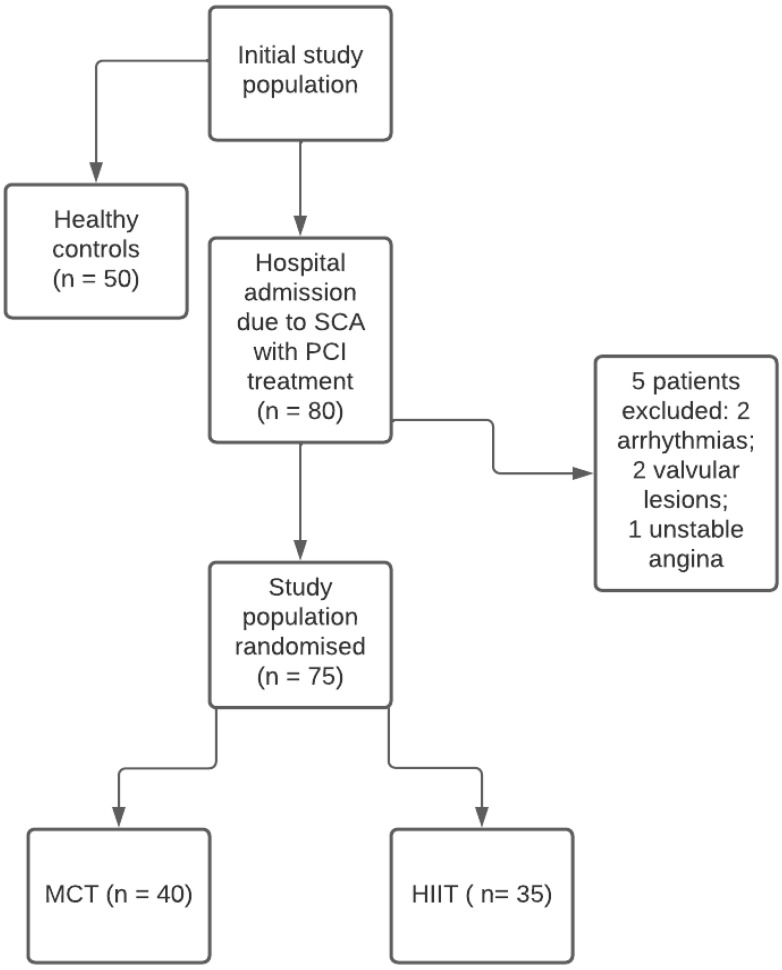
Flowchart. Diagram demonstrating the study protocol. ACS = acute coronary syndrome; PCI = Percutaneous Coronary Intervention; MCT = moderate-intensity continuous training; HIIT = high-intensity interval training.

**Figure 2 medicina-58-00453-f002:**
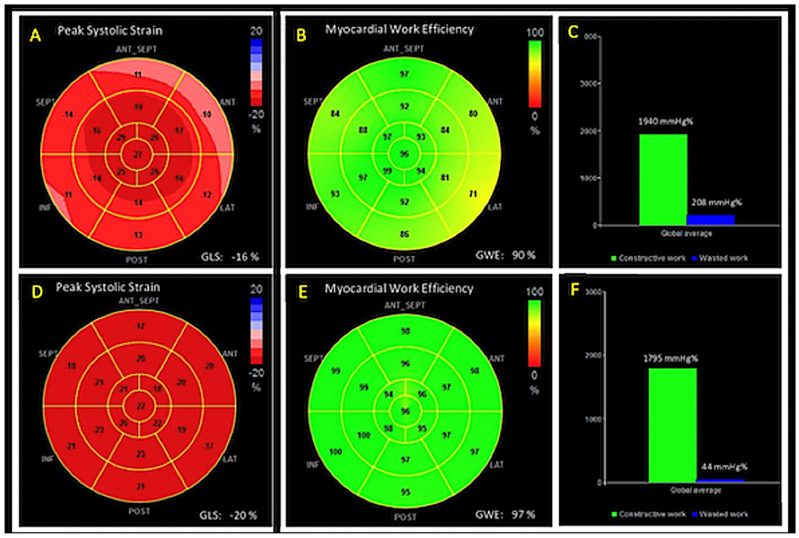
Panels (**A**–**C**): 17-segment bull’s-eye representation of LV strain (**A**), and myocardial work efficiency (**B**) and wasted work (**C**) in a patient after ACS performing rehabilitation. Both myocardial deformation (GLS −16%) and efficiency (90%) were moderately impaired. Panels (**D**–**F**): 17-segment bull’s-eye representation of LV strain (**D**), and myocardial work efficiency (**E**) and wasted work (**F**) in a healthy control. Both myocardial deformation (GLS −20%) and efficiency (97%) were within normal limits.

**Table 1 medicina-58-00453-t001:** Demographics and clinical features of the ischemic patients and healthy controls.

Variable	Ischemic Patients (*n* = 75)	Controls (*n* = 50)	*p*-Value
Male sex (%)	43 (57.5)	30 (60)	NS
Age (years)	62.3 ± 8.3	59.3 ± 15.4	NS
BSA (m^2^)	1.87 ± 0.17	1.85 ± 0.15	NS
BMI (kg/m^2^)	27.9 ± 3.6	25.2 ± 3.4	NS
Systolic blood pressure (mmHg)	133.5 ± 6.4	125.4 ± 5.8	<0.01
Diastolic blood pressure (mmHg)	85.4 ± 12.2	78.3 ± 9.2	<0.05
Heart rate (b/m)	78.4 ± 12.3	76.4 ± 11.4	NS
Arterial hypertension (%)	19 (25.5)	-	-
Diabetes mellitus (%)	31 (41.2)	-	-
Smoking or history of smoking (%)	33 (44.4)	-	-
Hyperlipidemia (%)	42 (56.6)	-	-
Beta-blockers (%)	65 (86.4)	-	-
ACE inhibitors or ARB (%)	58 (77.3)	-	-
Aldosterone receptor antagonists (%)	14 (18.5)	-	-
Antiplatelet agents/oral anticoagulants (%)	75 (100)	-	-
Diuretics (%)	27 (35.8)	-	-

Data are expressed as absolute number (%) or mean ± SD. BMI = body mass index; BSA: body surface area; b/m beat per minutes. ACE: angiotensin-converting enzyme; ARB: angiotensin II receptor blockers. NS = not statistically significant.

**Table 2 medicina-58-00453-t002:** Baseline echocardiographic measurements in the ischemic patients and control group.

Variable	Ischemic Patients	Controls	*p*-Value
	Ischemic Patients (*n* = 75)	Controls (*n* = 50)	
IVSd (mm)	11.8 ± 2.8	8.1 ± 2.3	<0.01
PWd (mm)	10.8 ± 1.6	7.7 ± 2.1	<0.01
LVEDD (mm)	53.3 ± 4.2	48.4 ± 5.6	<0.01
LVESD (mm)	47.4 ± 5.2	34.6 ± 4.8	<0.01
LV mass index (g/m^2^)	142.5 ± 22.3	80.5 ± 14.4	<0.0001
LVEDV index (mL/m^2^)	69.4 ± 11.4	60.6 ± 8.3	<0.01
LVESV index (mL/m^2^)	37.5 ± 12.3	24.5 ± 9.9	<0.001
Biplane LVEF (%)	47.1 ± 6.1	56.3 ± 5.5	<0.0001
LV GLS (%)	−12.8 ± 2.8	−21.4 ± 4.4	<0.0001
Myocardial Work Efficiency (%)	82.1 ± 3.3	94.4 ± 4.1	<0.001
Myocardial Wasted Work (%)	17.9 ± 4.4	5.8 ± 3.8	<0.0001
Mitral E velocity (m/s)	0.9 ± 0.6	0.8 ± 0.4	NS
Mitral A velocity (m/s)	0.7 ± 0.4	0.7 ± 0.3	NS
E/A ratio	1.3 ± 0.4	1.1 ± 0.4	NS
Mitral septal E’ velocity (m/s)	0.13 ± 0.05	0.16 ± 0.05	<0.01
Mitral lateral E’ velocity (m/s)	0.14 ± 0.04	0.18 ± 0.03	<0.01
E/e’ratio	9.9 ± 3.1	4.9 ± 2.8	<0.001
Aortic root diameter (mm)	34.3 ± 4.8	30.2 ± 3.2	<0.01
LAVI (mL/m^2^)	33.4 ± 4.1	28.3 ± 5.1	<0.01
LA Strain (%)	43.3 ± 4.1	53.8 ± 5.1	<0.001
PASP (mmHg)	37.5 ± 7.8	21.3 ± 2.9	<0.001
TAPSE (mm)	18.5 ± 3.3	24.5 ± 3.8	<0.01
Tricuspid S’ velocity (cm/s)	11.3 ± 2.2	14.4 ± 3.1	<0.01
B-lines (median and IQR)	1.45 (0–35)	0.70 (0–25)	<0.001

Data are expressed as mean ± SD or median (interquartile range). IVSd = inter-ventricular septum thickness at end diastole; PWd = posterior wall thickness at end diastole; LVEDD = left ventricular end diastolic diameter; LVESD = left ventricular end systolic diameter; LV = left ventricle; LVEF = left ventricular ejection fraction; LV GLS = left ventricular global longitudinal strain; LA = left atrial; LAVI = left atrial volume index; PASP = systolic pulmonary artery pressure; TAPSE = tricuspid annular plane systolic excursion. NS = not statistically significant.

**Table 3 medicina-58-00453-t003:** Resting echocardiographic and lung ultrasound evaluation in the overall population of ischemic patients at baseline and after cardiac rehabilitation.

Variable	Baseline	After Rehabilitation	*p*-Value
	Ischemic Patients (*n* = 75)		
	Baseline	After Rehabilitation	
IVSd (mm)	11.8 ± 2.8	10.9 ± 2.3	NS
PWd (mm)	10.8 ± 1.6	10.2 ± 2.1	NS
LVEDD (mm)	53.3 ± 4.2	54.4 ± 6.6	NS
LVESD (mm)	47.4 ± 5.2	41.6 ± 5.8	<0.01
LV mass index (g/m^2^)	142.5 ± 22.3	136.5 ± 13.4	NS
LVEDV index (mL/m^2^)	69.4 ± 11.4	70.6 ± 8.3	NS
LVESV index (mL/m^2^)	37.5 ± 12.3	33.5 ± 9.6	<0.01
Biplane LVEF (%)	47.1 ± 6.1	52.3 ± 5.5	<0.01
LV GLS (%)	−12.8 ± 2.8	−16.4 ± 4.3	<0.001
Myocardial Work Efficiency (%)	82.1 ± 3.3	89.4 ± 4.1	< 0.001
Myocardial Wasted Work (%)	17.9 ± 4.4	10.6 ± 3.3	<0.0001
Mitral E velocity (m/s)	0.9 ± 0.6	0.7 ± 0.4	NS
Mitral A velocity (m/s)	0.7 ± 0.4	0.9 ± 0.3	NS
E/A ratio	1.3 ± 0.4	0.7 ± 0.5	<0.01
Mitral septal E’ velocity (m/s)	0.13 ± 0.05	0.15 ± 0.05	<0.01
Mitral lateral E’ velocity (m/s)	0.14 ± 0.04	0.17 ± 0.03	<0.01
E/e’ratio	9.9 ± 3.1	4.4 ± 2.8	<0.001
Aortic root diameter (mm)	34.3 ± 4.8	34.2 ± 3.2	NS
LAVI (mL/m^2^)	33.4 ± 4.1	30.3.3 ± 5.1	<0.01
LA Strain (%)	43.3 ± 4.1	47.3 ± 5.4	<0.01
PASP (mmHg)	37.5 ± 7.8	30.3 ± 2.9	<0.01
TAPSE (mm)	18.5 ± 3.3	19.5 ± 3.8	NS
Tricuspid S’ velocity (cm/s)	11.3 ± 2.2	12.4 ± 3.1	NS
B-lines (median and IQR)	1.45 (0–35)	0.90 (0–25)	<0.001

Data are expressed as mean ± SD or median (interquartile range). IVSd = inter-ventricular septum thickness at end diastole; PWd = posterior wall thickness at end diastole; LVEDD = left ventricular end diastolic diameter; LVESD = left ventricular end systolic diameter; LV = left ventricle; LVEF = left ventricular ejection fraction; LV GLS = left ventricular global longitudinal strain; LA = left atrial; LAVI = left atrial volume index; PASP = systolic pulmonary artery pressure; TAPSE = tricuspid annular plane systolic excursion. NS = not statistically significant.

**Table 4 medicina-58-00453-t004:** Standard and strain echocardiographic measurements in ischemic patients after two different protocols of cardiac rehabilitation.

Variable	HIIT	MCT	*p*-Value
	HIIT (35 pts)	MCT (40 pts)	
IVSd (mm)	10.5 ± 2.9	11.1 ± 2.3	NS
PWd (mm)	10.2 ± 1.6	10.1 ± 2.1	NS
LVEDD (mm)	53.9 ± 4.2	54.8 ± 6.6	NS
LVESD (mm)	40.4 ± 6.2	43.6 ± 5.6	<0.05
LV mass index (g/m^2^)	133.5 ± 22.5	137.5 ± 12.4	NS
LVEDV index (mL/m^2^)	68.6 ± 11.2	71.2 ± 7.3	NS
LVESV index (mL/m^2^)	31.5 ± 11.3	35.6 ± 8.6	<0.01
Biplane LVEF (%)	53.1 ± 6.4	52.3 ± 5.4	<0.01
LV GLS (%)	−17.8 ± 3.8	−15.4 ± 4.3	<0.01
Myocardial Work Efficiency (%)	91.1 ± 3.3	87.4 ± 4.1	<0.01
Myocardial Wasted Work (%)	9.9 ± 4.4	12.6 ± 3.3	<0.01
Mitral E velocity (m/s)	0.8 ± 0.6	0.7 ± 0.4	NS
Mitral A velocity (m/s)	0.9 ± 0.4	0.8 ± 0.3	NS
E/A ratio	0.88 ± 0.4	0.87.1 ± 0.5	NS
Mitral septal E’ velocity (m/s)	0.16 ± 0.05	0.14 ± 0.05	<0.01
Mitral lateral E’ velocity (m/s)	0.18 ± 0.04	0.15 ± 0.03	<0.01
E/e’ratio	3.4 ± 3.1	6.4 ± 2.8	<0.01
Aortic root diameter (mm)	34.2 ± 3.8	34.2 ± 3.1	NS
LAVI (mL/m^2^)	29.4 ± 3.8	32.3 ± 4.1	<0.01
LA Strain (%)	49.2 ± 3.9	43.6 ± 4.1	<0.01
PASP (mmHg)	28.5 ± 7.8	33.3 ± 2.9	<0.01
TAPSE (mm)	19.5 ± 3.3	19.2 ± 3.7	NS
Tricuspid S’ velocity (cm/s)	12.7 ± 2.2	12.2 ± 3.1	NS
B-lines (median and IQR)	0.87 (0–27)	0.91 (1–25)	NS

Data are expressed as mean ± SD or median (interquartile range). IVSd = inter-ventricular septum thickness at end diastole; PWd = posterior wall thickness at end diastole; LVEDD = left ventricular end diastolic diameter; LVESD = left ventricular end systolic diameter; LV = left ventricle; LVEF = left ventricular ejection fraction; LV GLS = left ventricular global longitudinal strain; LA = left atrial; LAVI = left atrial volume index; PASP = systolic pulmonary artery pressure; TAPSE = tricuspid annular plane systolic excursion. NS = not statistically significant.

**Table 5 medicina-58-00453-t005:** Univariable analysis: Correlations between resting LV and LA echo indexes and functional parameters during effort in CR patients.

Variable		R	*p* Value
LV EF	VO_2_ Peak	0.21	NS
	LV E/e’ during ESE	−0.31	<0.05
	B lines during ESE	−0.2	NS
LV GLS	VO_2_ Peak	−0.40	<0.01
	LV E/e’ during ESE	0.33	<0.05
	B lines during ESE	0.23	NS
LA Strain	VO_2_ Peak	0.36	<0.01
	LV E/e’ during ESE	−0.36	<0.01
	B lines during ESE	−0.28	<0.05
LV MWE	VO_2_ Peak	0.52	<0.001
	LV E/e’ during ESE	−0.49	<0.001
	B lines during ESE	−0.39	<0.01

LV FE: left ventricular ejection fraction; GLS = global longitudinal strain; MWE: myocardial work efficiency. ESE = exercise stress echocardiography.

## Data Availability

The data presented in this study are available on request from the corresponding author.

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
