# Peer review of "Effects of High Intensity Interval Training Rehabilitation Protocol after an Acute Coronary Syndrome on Myocardial Work and Atrial Strain"

_medicina, 2022, doi:10.3390/medicina58030453_

Round 1

Reviewer 1 Report

In the present report, the authors aimed to investigate the effects of two different cardiac rehabilitation protocols HIIT and MCT, on cardiac remodeling and deformation indices in patients with acute coronary syndrome. The current study is clear, concise, presented in a well-structured manner, and relevant for the field of cardiovascular diseases.

Minor comments:

  1. Captions of the tables (Table 2, 3, and 4) need to be consistent since the evaluated parameters are the same between the tables.
  2. Since the difference between HIIT and MCT has been shown in Table 4, the current reviewer is curious to know the purpose of the result in Table 3 after cardiac rehabilitation for all ischemic patients.
  3. The abstract can be improved in the revised manuscript before publication. The author may consider focusing on the difference between HIIT and MCT in the abstract section.

Author Response

REVIEWER 1

In the present report, the authors aimed to investigate the effects of two different cardiac rehabilitation protocols HIIT and MCT, on cardiac remodeling and deformation indices in patients with acute coronary syndrome. The current study is clear, concise, presented in a well-structured manner, and relevant for the field of cardiovascular diseases.

Minor comments:

  1. Captions of the tables (Table 2, 3, and 4) need to be consistent since the evaluated parameters are the same between the tables.

As properly suggested, the 3 legends were corrected and homogenized

  1. Since the difference between HIIT and MCT has been shown in Table 4, the current reviewer is curious to know the purpose of the result in Table 3 after cardiac rehabilitation for all ischemic patients.

We thank the reviewer for this comment. Our only aim in table 3 was to show the overall effects of cardiac rehabilitation in the whole population of ischemic patients.

  1. The abstract can be improved in the revised manuscript before publication. The author may consider focusing on the difference between HIIT and MCT in the abstract section.

As properly suggested, we included in the new abstract a sentence about the two different protocols.

Reviewer 2 Report

Authors investigated the significance of high intensity interval training (HIIT) cardiac rehabilitation protocol after an acute coronary syndrome with advanced echocardiographic parameters of myocardial function. For a greater functional performance and higher quality of life, cardiac rehabilitation is quite important. Since its protocol needs to be updated, this study can be a cutting edge of new protocol. However, there are a lot of concerns to publish. Please answer the reviewer’s concerns.

Major

  1. Authors must declare that this study bases on CONSORT guidelines as a randomized controlled trial. According to those guidelines, authors should show the trial design, study settings (settings and locations where the data were collected), stopping guidelines (especially, safety concern), randomization (Method used to generate the random allocation sequence), blinding, and participant flow at least. In addition, did authors enroll the protocol to any institution? If yes, please identify the enroll number.
  2. The explanation of Table 1 is insufficient. Authors should explain the characteristics of participants.

Minor

  1. Did authors show Figure 1 D-E-F as control? If yes, please add the explanation in the legend.
  2. In Table 1, authors should show the absolute number before the percentage between “Arterial hypertension” and “Diuretics”. In addition, please describe abbreviation for ACE and ARB.
  3. Is the percentage of ACE-I or ARB correct (not ARB “blocker”)? The reviewer thinks this is low as the secondary prevention.
  4. Were there any differences between severity (e.g. SYNTAX score or the number of branches) as a subgroup analysis?

Author Response

REVIEWER 2

Authors investigated the significance of high intensity interval training (HIIT) cardiac rehabilitation protocol after an acute coronary syndrome with advanced echocardiographic parameters of myocardial function. For a greater functional performance and higher quality of life, cardiac rehabilitation is quite important. Since its protocol needs to be updated, this study can be a cutting edge of new protocol. However, there are a lot of concerns to publish. Please answer the reviewer’s concerns.

Major

  1. Authors must declare that this study bases on CONSORT guidelines as a randomized controlled trial. According to those guidelines, authors should show the trial design, study settings (settings and locations where the data were collected), stopping guidelines (especially, safety concern), randomization (Method used to generate the random allocation sequence), blinding, and participant flow at least. In addition, did authors enroll the protocol to any institution? If yes, please identify the enroll number.

We really thank the reviewer for this comment. As properly suggested, in different parts of the manuscript we specified all the requested points, in accordance with CONSORT guidelines (page 4, line 80; page 5 , line 96, and lines 105-107 and 113). In addition, we introduced as suggested also a new Figure 1 with the flow-chart diagram, depicting the study protocol.    

  1. The explanation of Table 1 is insufficient. Authors should explain the characteristics of participants.

As properly suggested, we introduced at page 6 , lines 175-182, a comment about the characteristics of our patients.

Minor

  1. Did authors show Figure 1 D-E-F as control? If yes, please add the explanation in the legend.

As suggested, we specified this point.

  1. In Table 1, authors should show the absolute number before the percentage between “Arterial hypertension” and “Diuretics”. In addition, please describe abbreviation for ACE and ARB.

We described now these abbreviations

  1. Is the percentage of ACE-I or ARB correct (not ARB “blocker”)? The reviewer thinks this is low as the secondary prevention.

We thanks the author for this point, we comment on it at page 6, lines 180-182.

  1. Were there any differences between severity (e.g. SYNTAX score or the number of branches) as a subgroup analysis?

We did non find any differences in these subgroups, since the main determinants were functiomnal indexes like ejection fraction or strain.

Round 2

Reviewer 2 Report

Authors answered and modified correctly. The reviewer will accept in present form.